

# Serotypes, virulence factors and multilocus sequence typing of *Glaesserella parasuis* from diseased pigs in Taiwan

Ching-Fen Wu[1], Chia-Yu Hsu[1], Chi-Chung Chou[2], Chao-Min Wang[1], Szu-Wei Huang[1] and Hung-Chih Kuo[1]

[1] Department of Veterinary Medicine, National Chiayi University, Chiayi City, Taiwan
[2] Department of Veterinary Medicine, National Chung Hsing University, Taichung, Taichung, Taiwan

## ABSTRACT

**Background.** *Glaesserella parasuis* (*G. parasuis*) belongs to the normal microbiota of the upper respiratory tract in the swine, but virulent strains can cause systemic infections commonly known as Glässer's disease that leads to significant economic loss in the swine industry. Fifteen serotypes of *G. parasuis* have been classified by gel immunodiffusion test while the molecular serotyping based on variation within the capsule loci have further improved the serotype determination of unidentified field strains. Serovar has been commonly used as an indicator of virulence; however, virulence can be significantly differ in the field isolates with the same serotype. To date, investigations of *G. parasuis* isolated in Taiwan regarding antimicrobial resistance, serotypes, genotypes and virulence factors remain unclear.

**Methods.** A total of 276 *G.parasuis* field isolates were collected from 263 diseased pigs at the Animal Disease Diagnostic Center of National Chiayi University in Taiwan from January 2013 to July 2021. Putative virulence factors and serotypes of the isolates were identified by polymerase chain reaction (PCR) and antimicrobial susceptibility testing was performed by microbroth dilution assay. Additionally, the epidemiology of *G. parasuis* was characterized by multilocus sequence typing (MLST).

**Results.** Serotype 4 (33.3%) and 5 (21.4%) were the most prevalent, followed by nontypable isolates (15.9%), serotype 13 (9.4%), 12 (6.5%), 14 (6.2%), 7 (3.3%), 1 (1.8%), 9 (1.1%), 11 (0.7%) and 6 (0.4%). Nine out of 10 putative virulence factors showed high positive rates, including group 1 *vtaA* (100%), *fhuA* (80.4%), *hhdA* (98.6%), *hhdB* (96.0%), *sclB7* (99.6%), *sclB11* (94.9%), *nhaC* (98.2%), *HAPS_0254* (85.9%), and *cirA* (99.3%). According to the results of antimicrobial susceptibility testing, ceftiofur and florfenicol were highly susceptible (>90%). Notably, 68.8% isolates showed multidrug resistance. MLST revealed 16 new alleles and 67 new sequence types (STs). STs of these isolated *G. parasuis* strains were classified into three clonal complexes and 45 singletons by Based Upon Related Sequence Types (BURST) analysis. All the *G. parasuis* strains in PubMLST database, including strains from the diseased pigs in the study, were defined into two main clusters by Unweighted Pair Group Method with Arithmetic Mean (UPGMA). Most isolates in this study and virulent isolates from the database were mainly located in cluster 2, while cluster 1 included a high percentage of nasal isolates from asymptomatic carriers. In conclusion, this study provides current prevalence and antimicrobial susceptibility of *G. parasuis* in Taiwan, which can be used in clinical diagnosis and treatment of Glässer's disease.

Corresponding author
Hung-Chih Kuo,
hjkuo@mail.ncyu.edu.tw

bility testing, Multilocus sequence typing

## INTRODUCTION

The infectious disease induced by *Glaesserella parasuis* (*G. parasuis*) is known as Glässer's
disease, which causes polyserositis, arthritis and meningitis in 4 to 8-week-old pigs
(*Aragon, Segales & Tucker, 2019*). Fifteen serotypes of *G. parasuis* were identifieded by
gel immunodiffusion test (GID), but 10–40% of the isolates were non-typable with this
method (*Kielstein & Rapp-Gabrielson, 1992*). *Howell et al. (2013)* used a molecular method
to identify the serotypes of *G. parasuis* by analyzing the sequences of capsule locus, which
has improved the traditional serological typing method. Serotypes 4, 5, and 13 of *G. parasuis*
are the most prevalent in various countries (*Angen, Svensmark & Mittal, 2004*; *Cai et al.,
2005*; *Luppi et al., 2013*; *Marois et al., 2006*). While serotype 4 causes polyserositis, the
serotypes 5 and 13 infections lead to acute death of pigs (*Kielstein & Rapp-Gabrielson,
1992*). However, studies have also shown that there are high variations in the virulence
between serotypes, as well as between different strains of the same serotype (*Oliveira &
Pijoan, 2004*; *Lin et al., 2019*; *Macedo et al., 2021*).

Multilocus sequence typing (MLST) can be used for the evolution analysis of bacterial
population, epidemiological surveillance and identification of transmission chains.
Bacterial isolates are characterized by analyzing the seven loci of housekeeping genes
(*Enright & Spratt, 1999*). *Olvera, Cerdà-Cuéllar & Aragon (2006)* have analyzed 131
*G. parasuis* strains which formed six clusters. Most of the nasal isolates from asymptomatic
pigs belonged to the same cluster, while virulent strains that caused systemic infection
were in the other cluster. When 7 allele sequences were set into a phylogenetic tree by
using neighbor-joining (NJ) method, two clades were developed and most virulent strains
were in the same clade, which was similar to the result from Unweighted Pair Group
Method with Arithmetic Mean (UPGMA) (*Olvera, Cerdà-Cuéllar & Aragon, 2006*). In the
study of *Mullins et al. (2013)*, 127 strains of *G. parasuis* were divided into 116 genotypes. A
phylogenetic tree made by NJ method formed 2 clades, in which 94.6% of avirulent strains
were classified as clade 1 and 94.9% of the virulent strains were in clade 2 (*Mullins et al.,
2013*).

Besides vaccination for the prevention of Glässer's disease (*Liu et al., 2016*), the use of
antimicrobial agents have contributed to the treatment of Glässer's disease in pigs. Recent
studies have shown that antibiotic resistance of *G. parasuis* in multiple countries have been
increasing due to overuse of antimicrobial agents (*de la Fuente et al., 2007*; *Dayao et al.,
2014*; *Zhang et al., 2014*; *Miani et al., 2017*; *Nedbalcová, Zouharová & Sperling, 2017*; *Van et
al., 2020*). To date, the studies with regard to antimicrobial susceptibility testing, serotypes
and genotypes of *G. parasuis* in Taiwan are rare, and prevalence of virulence factors of
*G. parasuis* has not been investigated. Therefore, this study aims to investigate serotypes,
genotypes, virulence factors and antimicrobial susceptibility of *G. parasuis* strains isolated

from diseased pigs in Taiwan. Meanwhile, association of serotypes with virulence factors and antimicrobial resistance is explored.

## MATERIALS AND METHODS

### Vertebrate animal study

The study did not involve any animal experiment. The Institutional Animal Care and Use Committee (IACUC) of National Chiayi University did not deem it necessary to obtain formal approval for this study.

### Bacterial isolation and identification

From January 2013 to July 2021, a total of 276 *G. parasuis* isolates from 263 diseased pigs in 228 infected herds were submitted to the Animal Disease Diagnostic Center (ADDC), Department of Veterinary Medicine of National Chiayi University for necropsy. The collected samples included lungs, coelomic fluid (pericardial fluid, pleural fluid, and peritoneal fluid), synovial fluid and cerebrospinal fluid, which were cultured on the chocolate agar (BBL™; Becton Dickinson, Franklin Lakes, NJ, USA) and incubated under 5% $CO_2$ and 37 °C for 18 to 48 h. A single colony was collected for purification and amplication. Nucleic acid was extracted from the purified bacteria by using Taco™ DNA/RNA Extraction Kit (Taco, Taiwan). The primer pairs designed for the 16S rRNA of *G. parasuis* were amplified by PCR (*Oliveira, Galina & Pijoan, 2001*). After identification, bacteria were stored in brain heart infusion broth (Difco™; Becton Dickinson, Franklin Lakes, NJ, USA) containing 10% fetal bovine serum (FBS, Difco™; Becton Dickinson, Franklin Lakes, NJ, USA) and 20% glycerol at −80 °C. The isolates collected from the sites other than the lungs were classified into systemic sources.

### Serotyping and detection of virulence genes

The extracted nucleic acid was used for the identification of 15 serotypes of *G. parasuis* by PCR according to the studies of *Howell et al. (2015)* and *Jia et al. (2017)*. In addition, PCR was performed for the 10 virulence genes of *G. parasuis* including pericellular membrane transport proteins (*vtaA*, *fhuA*, *hhdA*, *hhdB*, *nhaC*, *HAPS_0254* and *cirA*), adhesion proteins (*sclB7* and *sclB11*) and phage-related gene (*Sack & Baltes, 2009*; *Wang et al., 2011*; *Olvera et al., 2012*).

### Antimicrobial susceptibility testing

According to the guidelines of the Clinical and Laboratory Standards Institute (CLSI), antimicrobial susceptibility testing was performed by micro-broth dilution method using haemophilus test medium broth (15 µg/mL NAD, 5 mg/mL yeast extract and 5% FBS). Fourteen antimicrobial agents for treating respiratory diseases in the swine and humans were selected, including penicillin G, amoxicillin, cefazolin, ceftiofur, oxytetracycline, doxycycline, tylosin, tylvalosin, clarithromycin, florfenicol, lincospectin, tiamulin, trimethoprim/sulfamethoxazole (TS) and enrofloxacin. Concentration of the antimicrobial agents ranged from 0.0625 to 2,048 µg/mL, and susceptibility was recorded along with the minimum inhibitory concentration (MIC) that inhibited 50% ($MIC_{50}$) and

90% ($MIC_{90}$) of the isolates. Breakpoints were identified based on the literature and CLSI guidance as mentioned elsewhere. For penicillin, ceftiofur, florfenicol, enrofloxacin and tiamulin, the breakpoints used were those recommended by *CLSI (2020)* for *Actinobacillus pleuropneumoniae* or *Pasteurella multocida* (*CLSI, 2020*). For clarithromycin and TS, the breakpoints used were recommended by *CLSI (2022)* for *Haemophilus influenzae* (*CLSI, 2022*). The breakpoint used for amoxicillin was previously recommended by *Schwarz et al. (2008)* for bacterial pathogens involved in porcine respiratory tract infections. As the standard methods and specific breakpoints for resistance of *G. parasuis* have not been established yet, all the values used were extrapolations. *Haemophilus influenzae* (ATCC 49247), *Escherichia coli* (ATCC25922), *Staphylococcus aureus* (ATCC29213), *Enterococcus faecalis* (ATCC29212) and *Pseudomonas aeruginosa* (ATCC27853) were used as quality control strains following CLSI guidance (*CLSI, 2022*).

## Multilocus sequence typing (MLST)

Nucleic acid extraction after bacteria purification was used for a molecular biological test according to the MLST method for *G. parasuis* established by *Mullins et al. (2013)*. The selected seven *G. parasuis* house-keeping genes were *atpD*, *infB*, *mdh*, *rpoB*, *6pgd*, *g3pd* and *frdB*. The amplified PCR products were submitted for bidirectional sequencing and the sequences were uploaded to BioNumerics® version7.6 (Applied Maths, USA) for analysis. The alleles and genotypes were then compared with PubMLST database. New allele sequences were uploaded to PubMLST for verification to obtain allele profiles and to define the sequence types (STs). Subsequently, the isolated strains were clustered according to the respective allele profiles by using BURST analysis of PubMLST. When five out of seven allele profiles were identical, these *G. parasuis* strains were identified as the same clonal complex (CC), otherwise the strains out of the CCs were singletons. Meanwhile, ancestral type of each CC was analyzed. Finally, multiple-comparison was performed for the allele profiles of all strains using BioNumerics® version 7.6, and the phylogenetic tree was developed by using the UPGMA algorithm.

## Statistical analysis

Association was determined by Chi-square test using Microsoft® Office Excel 2019. When $p < 0.05$, there is a significant association.

## RESULTS

### Bacterial isolation and identification

These 276 *G. parasuis* isolates were mainly collected from 4 anatomic isolation sites, including 188 isolates from lungs (68.1%; 188/276), 49 isolates from synovial fluid (17.8%; 49/276), 30 isolates from pericardial, pleural and peritoneal fluid samples (10.9%; 30/276) and 9 isolates from cerebrospinal fluid (3.3%; 9/276) (Table 1).

### Serotyping and detection of virulence genes

The identified serotypes included 5 isolates of serotype 1 (1.8%), 92 isolates of serotype 4 (33.3%), 59 isolates of serotype 5 (21.4%), 1 isolate of serotype 6 (0.4%), 9 isolates of
**Table 1    Serotypes of *G. parasuis* and their isolation sites.**

| Isolation site | Lung and bronchus | Synovial fluid | Pericardial, pleural and peritoneal fluid | Cerebrospinal fluid | Total |
|---|---|---|---|---|---|
| Serotype | | | | | |
| 1 | 2 | 1 | 1 | 1 | 5 |
| 4 | 71 | 11 | 8 | 2 | 92 |
| 5 | 36 | 16 | 6 | 1 | 59 |
| 6 | 1 | 0 | 0 | 0 | 1 |
| 7 | 6 | 1 | 2 | 0 | 9 |
| 9 | 1 | 2 | 0 | 0 | 3 |
| 11 | 0 | 0 | 1 | 1 | 2 |
| 12 | 13 | 4 | 1 | 0 | 18 |
| 13 | 16 | 7 | 2 | 1 | 26 |
| 14 | 12 | 2 | 1 | 2 | 17 |
| N[a] | 30 | 5 | 8 | 1 | 44 |
| Total | 188 | 49 | 30 | 9 | 276 |

**Notes.**
[a]unidentified

serotype 7 (3.3%), 3 isolates of serotype 9 (1.1%), 2 isolates of serotype 11 (0.7%), 18 isolates of serotype 12 (6.5%), 26 isolates of serotype 13 (9.4%) and 17 isolates of serotype 14 (6.2%), and 44 isolates were untypable (15.9%) (Table 1). Serotypes 4 and 5 were the main serotypes in the *G. parasuis* -infected pigs, followed by serotypes 13 and 12. This phenomenum was also seen in the analysis for distribution of serotypes per year. Notably, serotype 6 was first identified in 2021 (Table S1). Among the 276 isolates, we observed that a diseased pig could be infected with more than one serotype at the same time, and these *G. parasuis* strains containing different serotypes were isolated from different anatomic sites. Furthermore, a case (a herd) which submitted more than one diseased pig may exert different serotypes in different pigs. In this study, we found that a submitted case from a herd containing 2 diseased pig showed 4 different serotypes. Serotypes 4 and 5 were respectively isolated from pleural fluid and synovial fluid, in one pig, and serotypes 11 and 13 were respectively isolated from pleural fluid and synovial fluid in another one.

As shown in Table 1, serotype 4 was significantly associated with the isolation sites ($p < 0.05$). Additionally, the proportion of serotypes 5, 7, 12, 13, and 14 and the nontypable *G. parasuis* strains isolated from the lung was higher than those from the other isolation sites. The proportion of serotype 9 isolated from the synovial fluid was higher than that from the other isolation sites. Taken together, the results indicated that respiratory tract was the major isolation site of *G. parasuis*.

Identification of virulence factors and their prevalences out of 276 isolates were as follows: 276 isolates of group 1 *vtaA* gene (100%), 275 isolates of *sclB7* gene (99.6%), 274 isolates of *cirA* gene (99.3%), 272 isolates of *hhdA* gene (98.6%), 271 isolates of *nhaC* gene (98.2%), 265 isolates of *hhdB* gene (96.0%), 262 isolates of *sclB11* gene (94.9%), 237 isolates of *HAPS_0254* gene (85.9%), 222 isolates of *fhuA* gene (80.4%), and 23 isolates of
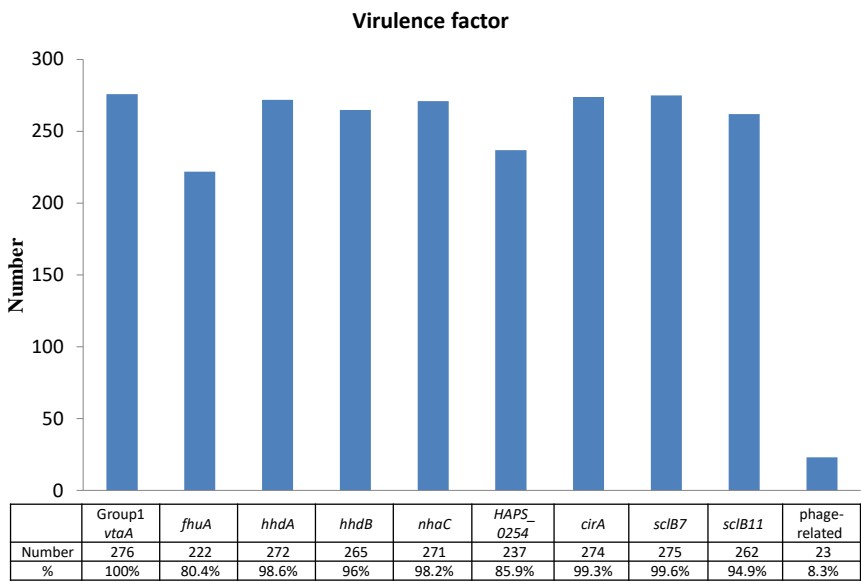

**Figure 1 Identification of virulence factors in the 276 isolates.** Percentage (%) was calculated as the number of isolates with detected virulence factors divided by the number of total isolates ($n = 276$).

phage-related gene (8.3%) (Fig. 1). Except for the phage-related gene, most of the isolates expressed the virulent genes were associated with pericellular membrane transport and adhesion simultaneously.

## Antimicrobial susceptibility testing

The proportion of susceptibility was 97.8% for ceftiofur, 94.0% for florfenicol, 90.6% for tiamulin, 73.9% for penicillin G, 37.5% for clarithromycin, 30.3% for enrofloxacin, 26.8% for amoxicillin and 18.0% for TS, indicating *G. parasuis* showed high susceptibility (>70%) to ceftiofur, florfenicol tilamulin and penicillin G. The susceptibility to amoxicillin, TS, clarithromycin and enrofloxacin was lower than 40%, meaning high resistance of *G. parasuis* to these antimicrobial agents. Among the 14 antimicrobial agents, the isolates showed the highest resistance rate to TS. The breakpoints of drug resistance for *G. parasuis* was not available for cefazolin, oxytetracycline, doxycycline, tylosin, tylvalosin and lincospectin, so that proportion of the drug resistance was not determined. Therefore, results were shown as $MIC_{50}$ and $MIC_{90}$ for these antimicrobial agents, which were 0.5 and 1 μg/mL for doxycycline, 0.5 and 2 μg/mL for cefazolin, 32 and 64 μg/mL for oxytetracycline, 32 and 256 μg/mL for tylvalosin, 32 and 512 μg/mL for lincospectin, and 64 and 512 μg/mL for tylosin (Table 2).

Antimicrobial resistance profile of *G. parasuis* showed that there were 28 isolates susceptible to all the antimicrobial agents (10.1%; 28/276), 235 isolates resistant to more than three classes of antimicrobial agents (68.8%; 190/276), and one isolate resistant to seven classes of antimicrobial agents (Table 3).

Association of antimicrobial resistance was also analyzed. It is worth noting that serotype 4 strains in this study had significantly higher resistance ($p < 0.01$) to both florfenicol and
**Table 2** **Minimum inhibitory concentration (MIC) distribution of *G. parasuis* strains isolated in this study (*n* = 276).**

| Antimicrobial agents | Number of isolations with MIC values (μg/mL) | | | | | | | | | | | | | | | | | MIC$_{50}$[a] (μg/mL) | MIC$_{90}$[a] (μg/mL) | R[b] (%) |
|---|---|---|---|---|---|---|---|---|---|---|---|---|---|---|---|---|---|---|---|---|
| | ≤ 0.0625 | 0.125 | 0.25 | 0.5 | 1 | 2 | 4 | 8 | 16 | 32 | 64 | 128 | 256 | 512 | 1,024 | 2,048 | >2,048 | | | |
| Penicillin G | | 104* | 61 | 39 \| | 19 | 7 | 7 | 15 | 4 | 5 | 2 | 8 | 5 | | | | | 0.25 | 8 | 26.1 |
| Amoxicillin | | 24* | 3 | 8 | 39 \| | 63 | 34 | 18 | 15 | 16 | 15 | 11 | 9 | 21 | | | | 4 | 256 | 73.2 |
| Cefazolin | | 61* | 50 | 69 | 50 | 25 | 7 | 2 | 1 | 2 | 0 | 1 | 8 | | | | | 0.5 | 2 | – |
| Ceftiofur | | 179* | 69 | 17 | 2 | 2 | 1 \| | 1 | 0 | 0 | 0 | 0 | 0 | 5 | | | | 0.125 | 0.5 | 2.2 |
| Oxytetracycline | | | | 47* | 12 | 7 | 6 | 7 | 40 | 83 | 68 | 4 | 1 | 1 | 0 | | | 32 | 64 | – |
| Doxycycline | | 65* | 42 | 85 | 83 | 1 | 0 | 0 | 0 | 0 | 0 | 0 | 0 | | | | | 0.5 | 1 | – |
| Tylosin | | | | 33* | 4 | 9 | 12 | 16 | 20 | 25 | 25 | 40 | 61 | 22 | 3 | 6 | | 64 | 512 | – |
| Clarithromycin | | 30* | 11 | 12 | 9 | 12 | 13 | 11 | 11 \| | 39 | 73 | 34 | 9 | 12 | | | | 32 | 128 | 62.5 |
| Tylvalosin | | | | 26* | 9 | 7 | 7 | 23 | 30 | 45 | 48 | 52 | 20 | 0 | 5 | 4 | | 32 | 256 | – |
| Florfenicol | 28* | 6 | 15 | 36 | 77 | 60 | 38 \| | 6 | 3 | 5 | 1 | 1 | | | | | | 1 | 4 | 6.0 |
| Lincospectin | | | | | 42 | 28 | 9 | 15 | 26 | 34 | 19 | 18 | 20 | 50 | 10 | 3 | 2 | 32 | 512 | – |
| Tiamulin | | 40* | 10 | 7 | 17 | 37 | 57 | 47 | 36 \| | 19 | 5 | 1 | 0 | | | | | 4 | 16 | 9.4 |
| Trimethoprim/ sulfamethoxazole | | | | 32* | 6 | 10 \| | 23 | 35 | 55 | 33 | 24 | 20 | 16 | 8 | 4 | 10 | | 16 | 256 | 82.0 |
| Enrofloxacin | | 60* | 13 | 8 \| | 65 | 66 | 23 | 25 | 11 | 2 | 3 | 0 | 0 | | | | | 1 | 8 | 69.7 |

**Notes.**

The dilution ranges tested for each antimicrobial agent are those contained within the white area. Breakpoints are indicated with vertical lines.

[a]MIC$_{50}$ and MIC$_{90}$ are the lowest concentration of antimicrobial agent capable of inhibiting the growth of 50% and 90% of the isolates, respectively.

[b]Percentage of the resistance.

*Asterisked numbers indicate the number of the isolates exhibiting MIC values equal to or lower than the concentrations of the test range.
**Table 3** Antimicrobial resistance profiles of *G. parasuis*.

| Number of isolates | Number of antimicrobial agents | Resistant patterns[a] |
|---|---|---|
| 28 | 0 | – |
| 5 | 1 | P |
| 1 | 1 | M |
| 7 | 1 | S |
| 1 | 1 | F |
| 2 | 2 | P + M |
| 1 | 2 | P + A |
| 18 | 2 | P + S |
| 4 | 2 | P + F |
| 7 | 2 | M + S |
| 2 | 2 | M + F |
| 10 | 2 | S + F |
| 7 | 3 | $P+M+S$ |
| 4 | 3 | $P+M+F$ |
| 2 | 3 | $P+A+S$ |
| 1 | 3 | P + PL + S |
| 29 | 3 | $P+S+F$ |
| 1 | 3 | $M+A+S$ |
| 14 | 3 | $M+S+F$ |
| 1 | 4 | $P+S+A+S$ |
| 97 | 4 | $P+M+S+F$ |
| 2 | 4 | P + PL + S + F |
| 2 | 4 | M + PL + S + F |
| 3 | 5 | $P+C+M+S+F$ |
| 1 | 5 | P + C + PL + S + F |
| 7 | 5 | $P+M+A+S+F$ |
| 14 | 5 | P + M + PL + S + F |
| 1 | 6 | $P+C+M+PL+S+F$ |
| 3 | 6 | $P+M+A+PL+S+F$ |
| 1 | 7 | $P+C+M+A+PL+S+F$ |

**Notes.**
[a]P, Penicillins (penicillin G, amoxicillin); C, Cephalosporin (ceftiofur); M, Macrolides (clarithromycin); A, Amphenicols (florfenicol); PL, Pleuromutilins (tiamilin); Sulfonamides: S, (trimethoprim-sulfamethoxazole); F, Fluoroquinolones (enrofloxacin).

tiamulin than the other serotype strains. The susceptibility of serotypes 1, 7, 9, 11, 13 and 14 was 100% for ceftiofur and florfenicol (Table S2). Additionally, the isolates collected from synovial fluid, in comparison with lung and bronchi, showed higher proportion of resistance to penicillin, amoxicillin, florfenicol, Tiamulin, TS and enrofloxacin; while all these isolates were susceptible to ceftiofur (Table S3).

## Multilocus sequence typing; MLST

Eighty-seven strains randomly selected from the isolated 276 *G. parasuis* strains according to the serotype proportion were selected for the MLST; namely from serotypes 4 ($n = 29$), 5 ($n = 18$), 13 ($n = 10$), 12 ($n = 6$), 14 ($n = 4$), 1 ($n = 2$), 7 ($n = 2$), 9 ($n = 2$), 6 ($n = 1$), 11 ($n = 1$) and untypable strains ($n = 12$). Sixteen new alleles of 7 house-keeping genes were found, including 1 allele for *atpD* gene, 3 for *infB* gene, 3 for *mdh* genes, 1 for *rpoB* gene, 2 for *6pgd* gene, 1 for *g3pd* gene, and 5 for *frdB* gene. STs of some strains could not match the PubMLST database; therefore, the new STs were uploaded to PubMLST for verification. Then the new allele profiles were submitted to PubMLST to define 67 new STs.

The 87 strains were divided into 3 CCs and 45 singletons by the BURST algorithm according to the similarity of the allele profiles. CC1 consisted of 14 STs and the ST593 was predicted as the ancestral type of this CC; CC2 included 14 STs and the ST570 was predicted as the ancestral type of this CC; CC3 comprised 2 STs and there was no predicted ancestral type (Fig. 2). There was no significant correlation between the CC and the strain isolation site, serotype, isolation sites or antimicrobial susceptibility. The main serotypes of CC1 included serotypes 4, 5, 12, 14 and 1 nontypable strain. Nine strains of serotype 5 were singletons. CC2 included serotypes 4, 12, 13, 14 and 4 nontypable strains, and most strains identified as serotype 13 were CC2.

The 757 strains of *G. parasuis* in the PubMLST database (including 87 strains of this study) were analyzed by using UPGMA algorithm and divided into two clusters (Appendix S1). There were 414 strains belonging to cluster 1 and 343 strains belonging to cluster 2. All the strains identified as serotypes 5 (30/30) and 14 (8/8) were in cluster 2, and serotype 4 (89.2%; 33/37), 12 (92.3%; 12/13) and 13 (93.8%; 15/16) were mainly distributed in cluster 2. Most strains of serotypes 1 (83.3%; 5/6), 2 (83.3%; 5/6), 7 (75%; 6/8), 9 (76.9%; 10/13) and 15 (75%; 6/8) were distributed in cluster 1. Serotype 6, 10, 11 and untypable strains were dispersed in the two clusters. Most of *G. parasuis* strains (77/83, 92.8%) isolated from healthy pig using nasal swabs were in cluster 1. Among the 87 strains collected in this study, 85 strains of *G. parasuis* were in cluster 2 and only 2 strains were in cluster 1.

## DISCUSSION

The main *G. parasuis* serotypes isolated in this study were serotypes 4 and 5, thus representing similar findings in other countries (*Angen, Svensmark & Mittal, 2004*; *Cai et al., 2005*; *Marois et al., 2006*; *Luppi et al., 2013*; *Ma et al., 2016*; *Lin et al., 2018*; *Schuwerk et al., 2020*; *Macedo et al., 2021*). In addition, these results are similar to the findings of *Cai et al. (2005)* that the strains identified as serotype 4 were prone to be isolated from the lungs, and the findings of *Luppi et al. (2013)* that isolation of serotype 5 was not correlated with the isolation sites (*Cai et al., 2005*; *Luppi et al., 2013*). *Macedo et al. (2021)* reported that distribution preference among the serotypes was not observed when comparing systemic and pulmonary isolates. Another recent study showed that isolates identified as serotypes 4 and 5, as well as other serotypes, had higher proportion of being collected from pulmonary isolation compared to the systemic isolation (*Espíndola et al., 2019*). Co-infecting pathogens are crucial factors for the distribution and replication of *G.*

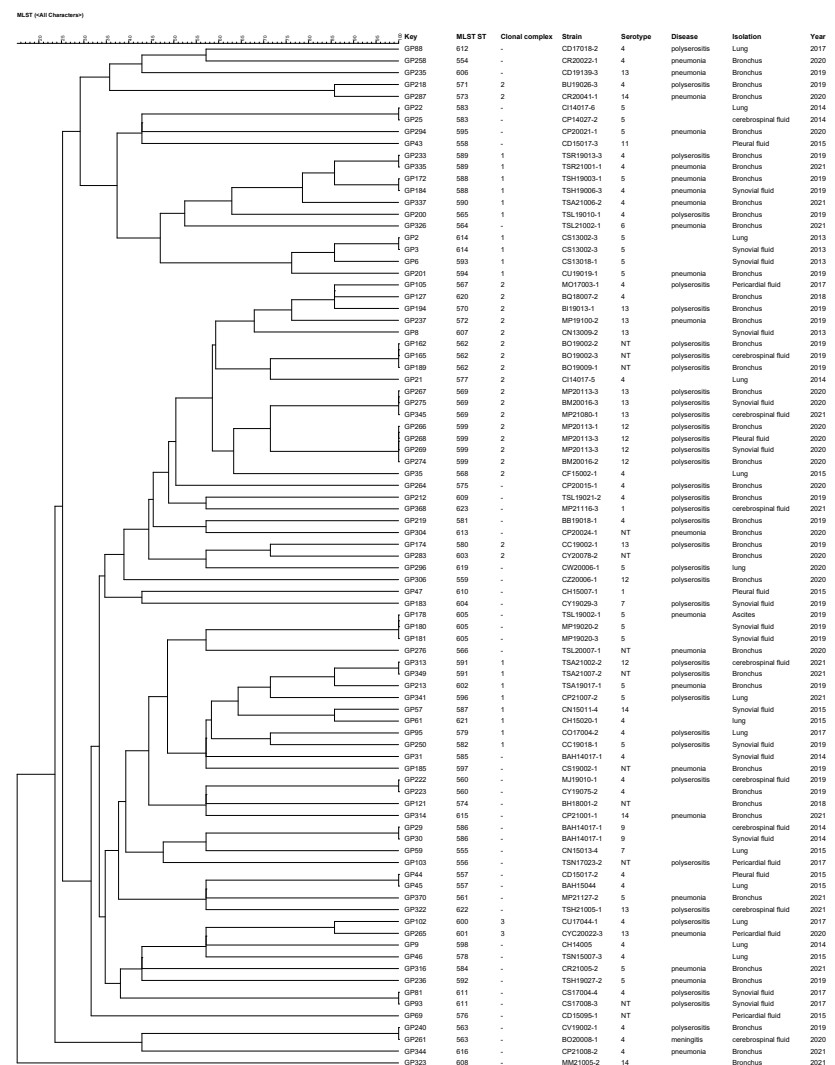

**Figure 2** UPGMA dendrogram constructed from the ST profiles of 87 *G. parasuis* isolates.

*parasuis*. In this study, regardless of serotypes, most of the diseased pigs were co-infected with porcine reproductive and respiratory syndrome virus (PRRSV) (232/276, 84.1%, Table S4), which may lead to reduction in phagocytosis of porcine alveolar macrophages (PAMs) (*Solano et al., 1998*). PRRSV-infected PAMs with phagocytosis impairment and the decrease in the bactericidal activity may contribute to lung damage favouring the development of secondary bacterial infections (*Renson et al., 2017*). PRRSV infection also induced higher levels of cytokines (IL-1 $\beta$, IL-18, IL-6 and TNF-$\alpha$) and bacterial loads of 11 bacterial species, including *G. parasuis*, in the lung (*Li et al., 2017*). Notably, there were 15.9% (44/276) nontypable *G. parasuis* strains. *Ma et al. (2016)* analyzed the full sequence of the capsule locus in the nontypable strains and found gene deficiency and unknown sequence in the locus of serotype specific region. In addition, the high proportion of

nontypable *G. parasuis* strains might be attributed to genetic differences in the *G. parasuis* populations between Europe and Asia, as the seroptying technique was developed using predominantly European strains.

The detection rate of *vtaA* gene in *G. parasuis* isolates was 100% in this study. The results from *Pavelko et al. (2016)* have also shown that the detection rate of *vtaA* gene in 23 strains of *G. parasuis* isolated from diseased pigs was 100%. In contrast, the detection rates of *vtaA* gene in healthy pigs was 4% (*Olvera et al., 2012*). The trimeric autotransporters encoded by *vtaA* gene have a high affinity for the host extracellular matrix protein to promote bacteria colonization. Group 1 *vtaA* gene is detected only in the virulent *G. parasuis* strains, which corresponded to our and other studies mentioned above (*Olvera et al., 2012*). In this study, group 1 *vtaA* gene was detected using the PCR for the *yadA*-like translocator domain of *vtaA*s, as *Olvera et al. (2012)* have perfomed. Recently, an improved PCR named leader sequence (LS)-PCR that has been reported by *Galofré-Milà et al. (2017)*, has been used in the studies for determining virulent and non-virulent isolates based on the *vta*-locus, specifically the extended signal peptide region (ESPR) (*Galofré-Milà et al., 2017*; *Schuwerk et al., 2020*; *Macedo et al., 2021*). *Galofré-Milà et al. (2017)* analyzed the strains that were detected using the described PCR for the *yadA* translocator domain of *vtaA*s with the new LS-PCR and found that all the strains determined as virulent strains in the LS-PCR were positive in the group 1-*vtaA* PCR, while all the strains determined as non-virulent strains were negative. In addition, *Schuwerk et al. (2020)* reported that the results of the LS-PCR were in good agreement with clinical metadata as well as with the results of the pathotyping PCR based on 10 different marker genes, a method for virulence prediction established by *Howell et al. (2017)*. LS-PCR is a good predictor used clinically for classifying virulence of the *G. parasuis* strains from healthy or diseased pigs. In this study, all the samples were collected from diseased pigs and were tested positive for group1 *vtaA* gene. According to the report of *Galofré-Milà et al. (2017)*, it is speculated that LS-PCR results for these samples would be virulent.

The detection rates of *fhuA*, *sclB7*, *sclB11*, *nhaC* and *HAPS_0254* were 80.4%, 99.6%, 94.9%, 98.2% and 85.9%, respectively. *Wang et al. (2011)* found that the detection rates of the these virulent genes in the virulent *G. parasuis* strains were significantly higher than in avirulent strains. In comparison, *Pavelko et al. (2016)* found that the detection rates of *fhuA*, *sclB7*, *sclB11*, *nhaC*, and *HAPS_0254* genes were mostly lower (except for *sclB11*) at 52.2%, 82.6%, 91.3%, 39.1% and 52.2%, respectively. In regard to the serotypes and their virulence, *Sack & Baltes (2009)* found that the putative hemolysin operon *hhdA* and *hhdB* were detected in the virulent strains of serotypes 5, 12, 13, 14 and 15, but not the avirulent strains of serotypes 3, 6, 7, 9 and 11. However, *Assavacheep, Assavacheep & Turni (2012)* found *hhdA* and *hhdB* genes in the avirulent strains of serotype 9 and 11. Moreover, the detection rate of *hhdA* gene (60%) was higher than the *hhdB* gene (40%) in *G. parasuis* isolated from the diseased pigs (*Assavacheep, Assavacheep & Turni, 2012*). *Pavelko et al. (2016)* also found that 56.5% and 47.8% of *G. parasuis* field strains were respectively detected the presence of *hhdA* and *hhdB* genes, and *hhdA* gene was detected in all the strains with the presence of *hhdB* gene. In this study, the positive rates of *hhdA* and *hhdB* genes were as high as 98.6% and 96.0%, respectively, and the *hhdA* gene was positive in all

*hhdB* gene positive strains, which corresponded to the findings of *Pavelko et al. (2016)*. In addition, we compared serotypes of all the collected strains with virulence factors (Table S5), and found that serotypes were not associated with virulence factors. Taken together, high detection rates of virulence factors were identified in the present study, and might be associated with the fact that the *G. parasuis* strains were isolated from diseased pigs. Comparative correlation analyses of virulence factors in *G. parasuis* isolated from healthy and diseased pigs are warranted.

The proportions of *β*-lactams resistant *G. parasuis* isolated from diseased pigs in Spain were 60% for penicillin G and 56.7% for ampicillin (*de la Fuente et al., 2007*). The proportions of 110 *G. parasuis* strains isolated in China resistant to ampicillin, amoxicillin and penicillin G were lower at 23.6%, 20.9% and 30%, respectively (*Zhang et al., 2014*). In another study, high proportions of *G. parasuis* were resistant to *β*-lactams antibiotics, including penicillin G (91.6%), ampicillin (100%), and amoxicillin (62.5%) (*Guo et al., 2012*). In Vietnam, there were high proportions of strains resistant to penicillin G, amoxicillin and cephalexin, which were 85.7%, 78.6% and 71.4%, respectively (*Van et al., 2020*). The isolated strains in the present study were highly susceptible to ceftiofur (97.8%). The result, in combination with the previous findings (*Aarestrup, Seyfarth & Angen, 2004*; *de la Fuente et al., 2007*; *Zhou et al., 2010*; *Nedbalcová & Kučerová, 2013*; *Dayao et al., 2014*; *Nedbalcová, Zouharová & Sperling, 2017*; *Brogden et al., 2018*), indicated that ceftiofur exerts significant therapeutic effects on pigs with Glässer's disease. Widespread of both *bla*$_{TEM-1}$ and plasmid pB1000-bearing *bla*$_{ROB-1}$ *β*-lactamase genes in *G. parasuis* strains is associated with elevated MICs of *β*-lactams (*San Millan et al., 2007*; *Guo et al., 2012*; *Zhao et al., 2018*). Additionally, 82% of the *G. parasuis* strains were resistant to TS in this study. Comparatively, the proportions of TS resistance in Spain, Australia and China were relatively lower at 53.3%, 40% and 44.5%, respectively (*Dayao et al., 2014*; *de la Fuente et al., 2007*; *Zhou et al., 2010*). *Zhao et al. (2018)* identified both *sul1* and *sul2*, the genes encoding for dihydropteroate synthases involved in sulfonamide resistance of Gram-negative bacteria and are associated with an integron system and a conjugative plasmid (*Vo et al., 2006*), may account for the resistance to TS. The proportion of *G. parasuis* resistance to enrofloxacin in the present study was similar to those in central and southern China (approximately 70%) but was lower than in Spain (20%) (*de la Fuente et al., 2007*; *Zhou et al., 2010*; *Zhang et al., 2014*). Mutations of quinolone resistance determining region mutations (QRDR) of *gyrA* and *parC* genes might be associated with fluoroquinolone resistance (*Zhao et al., 2018*).

Studies of antimicrobial resistance in other countries suggested that *G. parasuis* strains were mostly susceptible to florfenicol (*Aarestrup, Seyfarth & Angen, 2004*; *de la Fuente et al., 2007*; *Zhou et al., 2010*; *Nedbalcová & Kučerová, 2013*; *Dayao et al., 2014*; *Brogden et al., 2018*). The antimicrobial susceptibility results of *G. parasuis* to florfenicol showed that 94.0% of the strains were highly susceptible, indicating that in addition to ceftiofur, florfenicol could be used for treating Glässer's disease. *Zhao et al. (2018)* described 9% of *G. parasuis* strains studied carrying a novel small plasmid pHPSF1 bearing gene *floR*, a gene that accounts for florfenicol resistance. Furthermore, the proportion of *G. parasuis* resistant to tiamulin was 9.4% in this study, which was also comparable to data from other

countries including 0.9% in central China, 2.4% in Czech and somewhat higher at 40% in Spain (*de la Fuente et al., 2007*; *Zhou et al., 2010*; *Nedbalcová & Kučerová, 2013*).

Multi-drug resistance is defined as resistance to at least one agent in three or more antimicrobial categories (*Magiorakos et al., 2012*). In this study, the proportion of *G. parasuis* strains that were resistant to more than three classes of antimicrobial agents was 68.8%, meaning that multi-drug resistance was almost universal in Taiwan. Similar percentage (66.7%) has also been reported in Spain using 19 antimicrobial agents, and up to 23.3% of the tested *G. parasuis* strains were resistant to 8 antimicrobial agents simultaneously (*de la Fuente et al., 2007*). A recent report from Brazil has shown that 89.5% of tested strains were multi-drug resistant (*Silva et al., 2022*). Also, a report using 110 *G. parasuis* strains isolated from central China tested with 22 antimicrobial agents showed less severe multi-drug resistance percentage at 23.6% (*Zhou et al., 2010*), which is in accordance with a more recent preliminary study in Czech showing 20% of multi-drug resistance (*Nedbalcová, Zouharová & Sperling, 2017*).

Using the selected 87 *G. parasuis* strains obtained in this study, we have defined 16 new alleles. MLST analysis resulted in 67 new STs, which can be divided into 3 CCs and 45 singletons according to the similarity of allele profile. *G. parasuis* in the current PubMLST database (including the strains of this study) has 720 genotypes, showing the high heterogeneity and instability in *G. parasuis* strains. Previously, *Olvera, Cerdà-Cuéllar & Aragon (2006)* suggested that the *G. parasuis* clusters may be associated with their geographic locations. In the present study, while a part of the strains were singletons, the rest mainly belonged to CC1 or CC2. This phenomenon might be attributed to the fact that the sows of these pig farms were from the same source, leading to close strain evolutionary relationship. More samples from both healthy and diseased pigs are desirable to reveal true association between different backgrounds of the field strains and CC composition. *Mullins et al. (2013)* and *Wang et al. (2016)* showed that the avirulent strains isolated from healthy pigs were mainly in cluster 1, and the strains in cluster 2 were isolated mainly from pigs with pneumonia or systemic infection. In our study, 85 out of 87 strains isolated from diseased pigs were classified as cluster 2 (85/87, 97.7%), which corresponded well to the aforementioned literature.

The 87 strains in this study included serotypes 1, 4, 5, 6, 7, 9, 11, 12, 13, 14 and some nontypable strains. Serotype 1 strains ($n = 2$) were separately classified in cluster 1 and 2, serotype 11 strain ($n = 1$) was in cluster 1, and the other serotypes were all in cluster 2. It was reported that CA38-4 (serotype 12) and 373/03A (serotype 7) which did not cause death and polyserositis in the pigs were both in cluster 2 (*Aragon et al., 2010*), while *Mullins et al. (2013)* reported that the strains of serotype 1, 2 and 7 were in cluster 1, and those of serotypes 5 and 12 were in cluster 2. In addition, *Wang et al. (2016)* reported that cluster 1 contained the strains of serotypes 1, 2, 3, 4, 6, 7, 8, 9, 10 and 11, and cluster 2 contained the strains of serotypes 4, 5, 10, 12, 13, 14 and 15. As serotypes 4 and 10 were distributed in both clusters 1 and 2, the guinea pigs had been challenged by the strains of serotypes 4 and 10 from different clusters, and the result showed that strains YZ-4 (serotype 4) and H38 (serotype 10) from cluster 1 did not cause any clinical symptoms, whereas the guinea pigs challenged by H25 (serotype 4) and H367 (serotype 10) from cluster 2 showed depression,

arthritis, inappetence, and death (*Wang et al., 2016*). *Wang et al. (2016)* concludes that the strains in cluster 2 tend to be virulent, whereas the strains in cluster 1 were less virulent or avirulent. In the current study, both of the two serotype 1 strains in different clusters caused arthritis and polyserositis in pigs, and the serotypes were uncorrelated with the expression of virulence factors. Therefore, differentiation of the pathogenicity by serotype or cluster distribution may represent inappropriate tools and a challenge test is required to confirm the pathogenicity of the isolated *G. parasuis* strains.

## CONCLUSION

The prevalent serotypes of the *G. parasuis* strains isolated in the diseased pigs in Taiwan were mainly serotypes 4 and 5, followed by 12 and 13, which were consistent with the research in various countries. The MLST results showed new genotypes, demonstrating the diversity of *G. Parasuis* strains. Most of the strains isolated in this study were classified in cluster 2, which was in agreement with previous reports showing pathogenic strains are mainly located in cluster 2. Additionally, multi-drug resistant strains were common in this study, and ceftiofur and florfenicol could be the first choice of antimicrobial drugs for treatment. To sum up, this study updates the needed information on the molecular epidemiology of *G. parasuis* in Taiwan and highlights the resistant patterns of *G. parasuis* strains isolated from diseased pigs, providing guidances for on-site veterinarians with appropriate administration of antimicrobial agents.

### Funding
The authors received no funding for this work.

### Competing Interests
The authors declare there are no competing interests.

### Author Contributions
- Ching-Fen Wu analyzed the data, authored or reviewed drafts of the article, and approved the final draft.
- Chia-Yu Hsu performed the experiments, analyzed the data, prepared figures and/or tables, and approved the final draft.
- Chi-Chung Chou analyzed the data, authored or reviewed drafts of the article and approved the final draft.
- Chao-Min Wang analyzed the data, authored or reviewed drafts of the article, and approved the final draft.
- Szu-Wei Huang analyzed the data, authored or reviewed drafts of the article, and approved the final draft.
- Hung-Chih Kuo conceived and designed the experiments, analyzed the data, authored or reviewed drafts of the article, and approved the final draft.

## Animal Ethics

The following information was supplied relating to ethical approvals (i.e., approving body and any reference numbers):

The study did not involve any animal experiment. The Institutional Animal Care and Use Committee (IACUC) of National Chiayi University did not deem it necessary for this research group to obtain formal approval to conduct this study.

## DNA Deposition

The following information was supplied regarding the deposition of DNA sequences:

The 87 SCI MLST sequences are available in the Supplementary File and at https://pubmlst.org/bigsdb?db=pubmlst_hparasuis_isolates&page=query&prov_field1=f_country&prov_value1=Taiwan&submit=1.

## Data Availability

The raw measurements are available in the Supplementary File.

## Supplemental Information

Supplemental information for this article can be found online at http://dx.doi.org/10.7717/peerj.15823#supplemental-information.

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
