# Peer review of "Serotypes, virulence factors and multilocus sequence typing of Glaesserella parasuis from diseased pigs in Taiwan"

_PeerJ, doi:10.7717/peerj.15823_

## Round 0.1 · original submission · Major Revisions

Thank you for submitting your manuscript to PeerJ. I have now completed my evaluation of your manuscript. The reviewers recommend reconsideration of your manuscript following major revision. I invite you to resubmit your manuscript after addressing the comments below. When revising your manuscript, please consider all issues mentioned in the reviewers' comments carefully: please outline every change made in response to their comments and provide suitable rebuttals for any comments not addressed. Please note that your revised submission may need to be re-reviewed.

·

Basic reporting

Minor comments
Line 178-181: The sentence ‘Notably, we found 2 GP strains with four serotypes
Isolated....serotypes 5 and 13’ is not easy to understand, please rewrite it.
Line 191: It's better to change it to 'Taken together, the results...
Line 185: It's better to change it to ‘The results showed that serotype 4 was significantly correlated with the isolation sites’
Line 182-192: Please show the location of the results in manuscript, like Table 1.
Line 198: diseased pigs
Line 193-200: Please show the location of the results in manuscript.
Line 234: Figure 1: high-resolution fig should be provided.
Line 236: The main serotypes of CC1 included serotype 4, 5, 12, 14 and 1 untypable strain.
Line 252: 333 strains were isolated in this study, why 265 is presented in discussion part?
Line 284: serotypes?
Line 322-323: delete (2.4%) and (40%).
Line 325: ‘p’ Italic.
Line 338-339: rewrite the sentence.
Line376: polyserositis
Line 378: hallenge test? challenge
Line 380: rewrite the conclusion, this part is too long and needs to be more refined.
Table 1: The format of the table 1 should be consistent with the others. The lines in the table need to be bold.
Table S1: Add the full name of the abbreviation.

Experimental design

The study was well designed.

Validity of the findings

Conclusions are well stated.

Additional comments

No addional comments.

Reviewer 2 ·

Basic reporting

major concerns:
• Isolates used in this study: Did the authors ensure that the same strain isolated from different sites in one animal has not been examined several times? I propose to take into account only one isolate per animal or multiple isolates from one animal if they differ from each other, e.g. serotype or ST. Furthermore, it is not clear if several isolates derived from one outbreak represent a single strain in reality. The examination of several isolates, that represent a single strain in reality, should not affect the diversity of ST, but may have strong influence on the prevalence of serovar or virulence genes, overrepresenting strains where mulitple, identical isolates have been examined. I would suggest to ensure that only strains from different animals, farms and disease outbreaks (if occuring in a single farm, then just consider isolates whose isolation was at least 6 months apart) or different serotypes or ST are included in the study. If the authors do not have these metadata, please state that in your Material&Methods part, which would dramatically reduce the reliability of their results concerning statements on serovar and virulence gene prevalence.
• as it seems that your data are biased, I would propose to refrain from statistical analysis
• I do not expect that each isolate in the study is examined by MLST, as the message is clear: GP has a wide variety of STs occuring, making it an agent difficult to assess and comprehend. However, if the authors had properly sorted out duplicate strains, then a full examination of ST of prevalent strains in Taiwan could have been elaborated.
• regarding your citations: Please revise many of your citations, listet below under minor concerns
• Unfortunately, you did not use the leader sequence (LS) PCR published by Galofre-Mila et al., a tool that has been used in diverse studies showing good agreement with clinical data and is partly used in daily diagnostic. Please see at least the following citations and briefly discuss them in your manuscript (see Galofré Milà et al, 2017: A robust PCR for the differentiation of potential virulent strains of Haemophilus parasuis; Schuwerk et al., 2020: Serotyping and pathotyping of Glaesserella parasuis isolated 2012–2019 in Germany comparing different PCR-based methods; Macedo et al., 2021: Molecular characterization of Glaesserella parasuis strains isolated from North America, Europe and Asia by serotyping PCR and LS-PCR).
• please improve the written english text in grammatical terms. Online translaters such as deepl.com can be used for free and represent useful tools
• please at least refere or cite existing literature on GP virulence in Taiwan: Lin, 2019: Genotypic analyses and virulence characterization of Glaesserella parasuis isolates from Taiwan
• was antimicrobial susceptibility somehow connected with the serovar or the site of isolation? Please check this carefully and state a sentence in the results section.

Experimental design

no comment

Validity of the findings

no comment

Additional comments

minor concerns:
• line 26: investigations
• line 27: remain
• line 33: replace “then“ by “Additionally“
• line 35-36: replace “followed be the unidentified“ by “followed by nontypable isolates“
• line 66: “were non-typable with this method“
• line 67: delete biological
• line 74: Sack and Baltes is not appropriate. Please refer to another citation.
• line 78: please be more precise, who has described 7 house keeping genes in GP strains? The citation which has been inserted seems not to be appropriate
• line 80: which other cluster? There are 5 remaining. Please clarify.
• line 80: 7?
• line 87: Espindola et al did not perform vaccination. Please insert a more appropriate citation.
• line 95: delete “biological“
• line 111: „A single colony“
• line 118: Jia was only used to distinguish between serovars 5 and 12? Please clarify
• lines 122: please also cite the literature where these markers are originally described, as Pavelko et al also just used already published markers
• line 143: extraction
• line 154: why do you cite Olvera here, please clarify.
• line 182-184: Delete the two sentences.
• line 184-185: move this sentence to your material and methods section.
• lines 186-192: include odds ratios.
• line 191: results
• line 208-209: rewrite this sentence as it is not clear what you mean.
• line 211: replace “such“ by “so“
• lines 194-198: please show these data in a graph to make reading more comfortable
• line 198: diseased
• line 223: namely from serotype x (n=XXX), x (n=xxx) etc#
• line 231: introduce the abbreviation CC
• line 242 and 245: replace “1“ by “one“ (line 245: referring to the first “1“)
• line 251: delete both “the“
• lines 251-257: delete this paragraph as it provides no new information. Any new information of the present study should be stated in the results section
• line 258-259: rewrite: …serotype 4 and 5, thus representing similar findings in other countries
• line 259-260: cite more recent literature : Schuwerk 2020, Macedo 2021
• line 261: name again the isolation site to increase comprehensibility for the reader
• line 261-263: presumably highly biased results, thus do not overinterpret the odds. Please delete this sentence.
• line 267: 13.2% seems quite high in Asia comparing reports from europe. Please state in the manuscript that genetic differences in the GP populations between europe and asia is possible present. As the seroptying technique was developed using predominantly european strains, this might be an explanation
• line 270: confirm? you mean to serotype previous non-typable strains?
• line 271: replace “were“ by “was“
• line 278: why do you cite Pina et al again? Please find other studies that state that vtaA is predominantly present in virulent strains of GP. After the sentence, please insert a new paragraph
• line 281: replace “the“ by “in“
• line 284: “serotypes“
• line 290-292: please avoid writing about “gene positivity“. Rather state a phrase like “detected the presence of hhdA in xxx“
• line 295: please include this dataset in the supplemenary files and state in the results section oft he article that no link between serovar and presence of putative virulence genes was detected
• line 299: single reports exist, please see Howell et al., Schuwerk et al, Macedo
• line 310-311: avoid doubeling “ceftiofur“
• line 311-315: please rewrite the sentence to clarify why this literature was cited
• line 316: please rewrite this sentence
• lines 331-335: was multi-drug resistance now unique to Taiwan or not? Please clarify
• line 338: “n a“ ?
• lines 339-342: delete this sentence or include it in your conclusion section
• line 348: please separate the sentences with a dot after the citation
• line 376: “polyserositis“
• line 377: “differentiation“
• line 377-378: “… may represent inappopriate tools to examine the pathogenecity of isolated GP strains.“
• line 383-384: protection of vaccination was not evaluated in this study. Please delete this sentence
• line 386: replace “were“ by “was“
• line 386: “demonstrating“
• line 388: replace “literatures“ by “reports“
• line 390: “further research is needed“
• line 394: “describes“. Replace “features“ by “data“
• table 1 and 2: heading: delete “the“
• table S1: please explain abbreviations here, too

Reviewer 3 ·

Basic reporting

Literature references need to be updated. (More suggestions attached)

Experimental design

some methods need clarification (More suggestions attached)

Validity of the findings

Additional description of data is needed (More suggestions attached)

Annotated reviews are not available for download in order to protect the identity of reviewers who chose to remain anonymous.

---

## Round 0.2 · accepted · Accept

I have gone through the point-by-point rebuttal by the authors. According to the response letter, the authors have addressed all the reviewers' questions. Although we have invited the previous three reviewers to re-assess the revised manuscript, there is no response. I assume that they are happy with the revised version of the manuscript. Therefore, I endorse the acceptance of the manuscript.